# Super-resolution analysis of PACSIN2 and EHD2 at caveolae

**Tamako Nishimura[1], Shiro Suetsugu[1,2,3]**

**1** Division of Biological Science, Graduate School of Science and Technology, Nara Institute of Science and Technology, Ikoma, Japan, **2** Data Science Center, Nara Institute of Science and Technology, Ikoma, Japan, **3** Center for Digital Green-innovation, Nara Institute of Science and Technology, Ikoma, Japan

* suetsugu@bs.naist.jp

**Data Availability Statement:** All relevant data are within the manuscript and its Supporting Information files.

**Funding:** This work was supported by JSPS (https://www.jsps.go.jp/english/)(KAKENHI, JP 20H03252, JP20KK0341, JP21H05047) to SS and by JSPS (JP20K06625) to TN, and JST (https://

## Abstract

Caveolae are plasma membrane invaginations that play important roles in both endocytosis and membrane tension buffering. Typical caveolae have invaginated structures with a high-density caveolin assembly. Membrane sculpting proteins, including PACSIN2 and EHD2, are involved in caveolar biogenesis. PACSIN2 is an F-BAR domain-containing protein with a membrane sculpting ability that is essential for caveolar shaping. EHD2 is also localized at caveolae and involved in their stability. However, the spatial relationship between PACSIN2, EHD2, and caveolin has not yet been investigated. We observed the single-molecule localizations of PACSIN2 and EHD2 relative to caveolin-1 in three-dimensional space. The single-molecule localizations were grouped by their proximity localizations into the geometric structures of blobs. In caveolin-1 blobs, PACSIN2, EHD2, and caveolin-1 had overlapped spatial localizations. Interestingly, the mean centroid of the PACSIN2 F-BAR domain at the caveolin-1 blobs was closer to the plasma membrane than those of EHD2 and caveolin-1, suggesting that PACSIN2 is involved in connecting caveolae to the plasma membrane. Most of the blobs with volumes typical of caveolae had PACSIN2 and EHD2, in contrast to those with smaller volumes. Therefore, PACSIN2 and EHD2 are apparently localized at typically sized caveolae.

## Introduction

Caveolae are flask-shaped plasma membrane invaginations that are abundant in several cell types found in muscle, epithelial, and adipose tissues [1–3]. Caveolae play dual roles at the plasma membrane, as an endocytic apparatus and a membrane reservoir for buffering membrane tension. During endocytosis, the caveolar invagination is pinched off to form endocytic vesicles, while in tension buffering it is flattened to provide extra surface area to increase the membrane surface [1,4,5].

Caveolae are composed of a unique set of proteins and lipids. The caveolar membrane is rich in cholesterol, similar to the lipid rafts at the plasma membrane, where several receptors and signaling proteins are reportedly concentrated [6–9]. Caveolae are also a platform for signaling proteins that are regulated by the caveolar endocytic function. The structural caveolar proteins comprise caveolins and cavins [10,11]. Caveolin exists as three isoforms, and the caveolin-1 and

www.jst.go.jp/EN/) CREST (JPMJCR1863) to SS.
The funders had no role in study design, data
collection and analysis, decision to publish, or
preparation of the manuscript.

**Competing interests:** The author has declared that
no competing interests exist.

caveolin-3 amino acid sequences are almost identical [12,13]. Caveolin-1 is ubiquitously
expressed, while caveolin-3 is predominantly expressed in muscle. Mutations associated with dis-
eases such as muscular dystrophy have been identified in caveolin-3 [14,15], consistent with the
role of caveolae in the tension buffering of muscle cells [16]. There are four cavin isoforms, and
they are essential for caveolae [11,17–20]. Cavins associate with caveolins and generate the charac-
teristic striations on the caveolar surface, as observed by electron microscopy [21–24].

The endocytosis of caveolae is mediated by dynamin [25], as in clathrin-mediated endocytosis.
The invaginated membrane of clathrin-coated pits is mainly produced by BAR domain proteins
[26,27], which directly generate membrane curvatures and recruit structural proteins for mem-
brane remodeling, including dynamin and Wiskott–Aldrich syndrome family proteins [28].
Dynamin mediates the pinching of invaginations to form vesicles, in cooperation with the actin
cytoskeleton [29]. The BAR domains are divided into the BAR, N-BAR, and I-BAR domain sub-
families [30,31]. Among them, the F-BAR domain-containing protein PACSIN (Syndapin) is
involved in caveolae [32–34]. Three isoforms of PACSIN have been described. PACSIN3 is a
muscle-specific isoform, and its knockout results in caveolar biogenesis abnormalities [35]. PAC-
SIN2 is a ubiquitous isoform involved in caveolae formation and endocytosis [34]. PACSIN1 is
brain-specific, and its role in caveolae has not yet been clarified [34]. Importantly, PACSIN2 has
membrane deforming ability, which is altered by the cholesterol content of the membrane, imply-
ing the important role of PACSIN2 in caveolar homeostasis [36]. Indeed, PACSIN2 is stably local-
ized at caveolae, presumably at the neck of caveolar invaginations [33,34,37]. Furthermore,
PACSINs have NPF sequences that bind to the EHD2 protein, which is localized at and stabilizes
caveolae, presumably by mediating actin cytoskeleton anchoring [32,38,39]. Importantly, dyna-
min is recruited to caveolae only when PACSIN2 disappears, suggesting the regulation of dyna-
min binding to PACSIN2 during endocytosis [40].

Approximately 150 caveolin-1 molecules have been detected in mature caveolae [37,41].
Due to this abundance of caveolin-1, typical caveolae appear to have a certain caveolin-1 den-
sity, which can be measured by the nearest neighbor distance (NND) between caveolin-1s in
single-molecule localization microscopy (SMLM), a method that can determine the coordi-
nates of molecules at an accuracy equivalent to the protein size; that is, ~10 nm in the plane
parallel to the focal plane [42]. Using SMLM data projected onto a two-dimensional plane, the
membrane deformation can be monitored by the density estimations of caveolin-1 [42] or the
distances between caveolin-1 and caveolae-localized molecules [43]. Furthermore, by intro-
ducing a cylindrical lens, the localization depth measurement; that is, the three-dimensional
determination of the coordinates, could be achieved with an accuracy of ~50 nm in the depth
direction [44,45]. The three-dimensional coordinates of caveolin-1 localization can be grouped
into blob-shaped geometrical structures, which were previously classified into the typical
caveolae or other caveolin-1 clusters [46,47]. In this study, we used three-dimensional SMLM
to examine PACSIN2 and EHD2 localizations relative to caveolin-1. The PACSIN2 and EHD2
coordinates mostly overlapped with the caveolin-1 coordinates in the caveolin-1 blobs of
caveolar volume, which are thought to correspond to typical caveolae. However, PACSIN2
was ~5 nm closer to the plasma membrane than EHD2 and caveolin-1, suggesting its role in
connecting caveolae to the plasma membrane. Most of the blobs with volumes of typical caveo-
lae had PACSIN2 and EHD2, whereas those with smaller volumes did not. Therefore, PAC-
SIN2 and EHD2 are apparently localized at typically sized caveolae.

## Results and discussion

First, we observed the single-molecule localizations in antibody-stained HeLa cells. SMLM
uses total internal reflection to illuminate the fluorophore, and thus the observation is limited

to the plasma membrane neighboring the glass surface on which the cells attach. HeLa cells were labeled with antibodies against caveolin-1 and PACSIN2 or EHD2. Two kinds of antibodies for each protein were used, Caveolin-1 (7C8) + Caveolin-1 (3238), EHD2 (G-3) + Caveolin-1 (3238), EHD2 (11440-1-AP) + Caveolin-1 (7C8), PACSIN2 (SAB-1402538) + Caveolin-1 (3238), and PACSIN2 (Senju) + Caveolin-1 (7C8). These antibodies were visualized with secondary antibodies that were doubly labeled with Alexa 647 combined with Cy3 or Alexa 405. The activation of Alexa 405 or Cy3 by the excitation light was converted to the activation of Alexa 647, enabling the observation of the two labels at the same wavelength for Alexa 647. Therefore, the wavelength aberration was negligible between the observations of the two labels. The Alexa 647 signals that were associated with the activation Alexa 405 or Cy3 signals were considered for the analysis, to avoid the non-specific observations. To enable three-dimensional observations, a cylindrical lens was utilized to examine the signal depths by observing the deformation of the SMLM spherical signal according to the distance to the focal plane. The typical images reconstructed from the coordinates showed the proximity localizations of clustered signals of PACSIN2 and caveolin-1, as well as EHD2 and caveolin-1 (Fig 1).

The single-molecule localizations of caveolin-1 within an 80 nm distance from each other were grouped into four classes by the SuperResNet software, according to the number of signals, area, shape, and so on [46,47]. The clustering below 80 nm was considered to be reasonable because caveolae are typically 60–100 nm in diameter [10,48], and the number of observations of caveolin-1 was 4–15 within 100 nm of caveolae with this antibody labeling, according to our previous estimations for caveolar shapes [42]. The spatial distributions of these groups of signals were mostly spherical, and hence they are called blobs. The mapping of the blobs by t-distributed stochastic neighbor embedding (t-SNE) by Mahalanobis distance, cosine distance, and Euclidean distance resulted in the clear separation of the blobs, suggesting robust classification (Fig 2A). The number of blobs was consistent between observations (Fig 2B). The estimated size of caveolin-1 blobs in each class was also similar between observations, suggesting the accurate grouping of the signals (S1 Fig). As reported in the development of this method [47], the class 1 and class 2 blobs contained fewer caveolin-1 localizations and smaller volumes, while the class 3 blobs had volumes equivalent to those of typical caveolae (Fig 2C and 2D). The class 3 blobs typically contained abundant ~20 caveolin-1 signals. The volumes of the class 3 blobs were approximately ten times larger than those of the class 1 and 2 blobs, and were similar to the volumes of 60–100 nm spheres, strongly suggesting that class 3 corresponded to typical caveolae. The class 4 blobs contained superabundant localizations of ~100 signals, and thus might represent caveolae rosettes or clusters [49,50].

To estimate the colocalization of the two antibody labels, the spatial distribution of signals was estimated by comparing the distance between the centroids of the two antibody blobs to the standard deviations of the signal coordinates of blobs (Fig 2E). The colocalization was determined by the overlap of the spatial distributions of the two kinds of signals; i.e., the centroid distance below the sum of each standard deviation (Fig 2F). Approximately half of the class 1 and class 2 blobs of caveolin-1 had signals from another antibody against caveolin-1, PACSIN2, and EHD2 (Fig 2G). Within each class, the blobs with colocalization had larger volumes and signals than those without colocalization (S2 Fig). Therefore, the number of molecules to be detected is related to the volume of the blobs and the number of observed signals, suggesting that the class 1 and 2 blobs might have an insufficient number of molecules for colocalization detection or might not have colocalization. Such difficulty in analyzing small structures would probably result from the antibody labeling, where only a small fraction of the proteins can be visualized, as has been shown that 1–8 signals per caveola by electron microscopic analysis by using various caveolin-1 antibodies [51]. Most of the class 3 and class 4

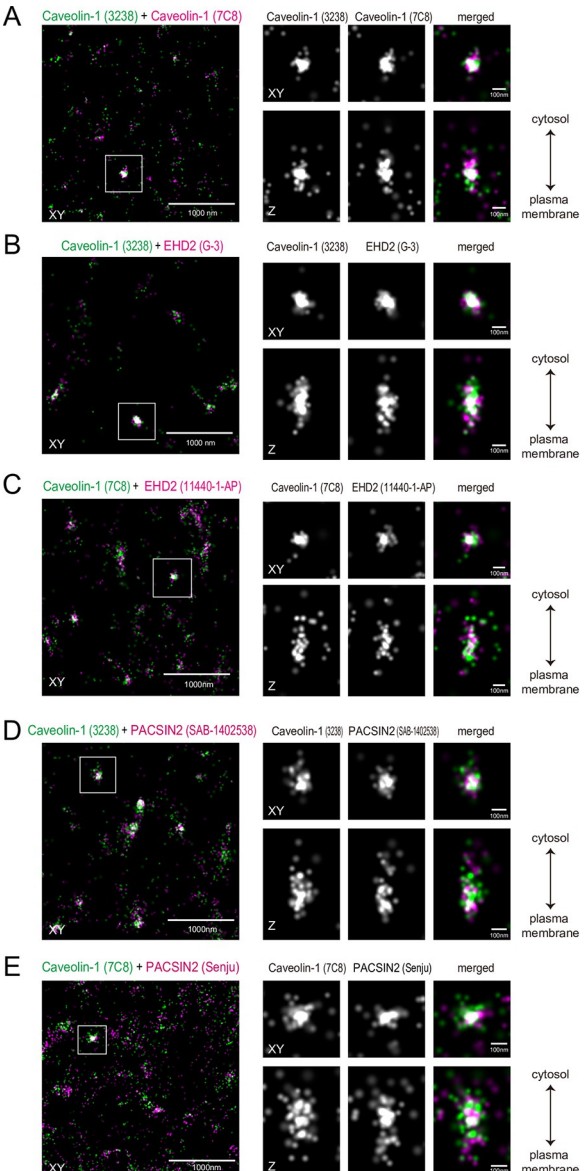

**Fig 1. SMLM analyses of PACSIN2, EHD2, and caveolin-1.** Representative reconstituted image of caveolin-1 (green) with those of PACSIN2 or EHD2 (magenta) in HeLa cells. The merged image of the focal plane (XY) is shown on the left. The box indicates the region for the enlarged images on the right, and contains one blob composed of caveolin-1 presumably corresponding to a caveola, with the projection to show the signal distribution to the Z-direction for each caveolin-1, PACSIN2, or EHD2 image. The combinations of antibodies are: (A) Caveolin-1 (7C8) + Caveolin-1 (3238), (B) Caveolin-1 (3238) + EHD2 (G-3), (C) Caveolin-1 (7C8) + EHD2 (11440-1-AP), (D) Caveolin-1 (3238) + PACSIN2 (SAB-1402538), and (E) Caveolin-1 (7C8) + PACSIN2 (Senju).

caveolin-1 blobs had the signals from another antibody to caveolin-1, PACSIN2, and EHD2, suggesting that caveolae with a typical size had PACSIN2 and EHD2.

We next examined the spatial distribution of the signals; i.e., the shape, of the class 3 blobs. The standard deviations in the XY plane, parallel to the glass surface or plasma membrane, and the Z direction in depth were similar for the caveolin-1 signals between various antibody combinations (Fig 3A and 3B). The standard deviations of the PACSIN2 and EHD2 signals

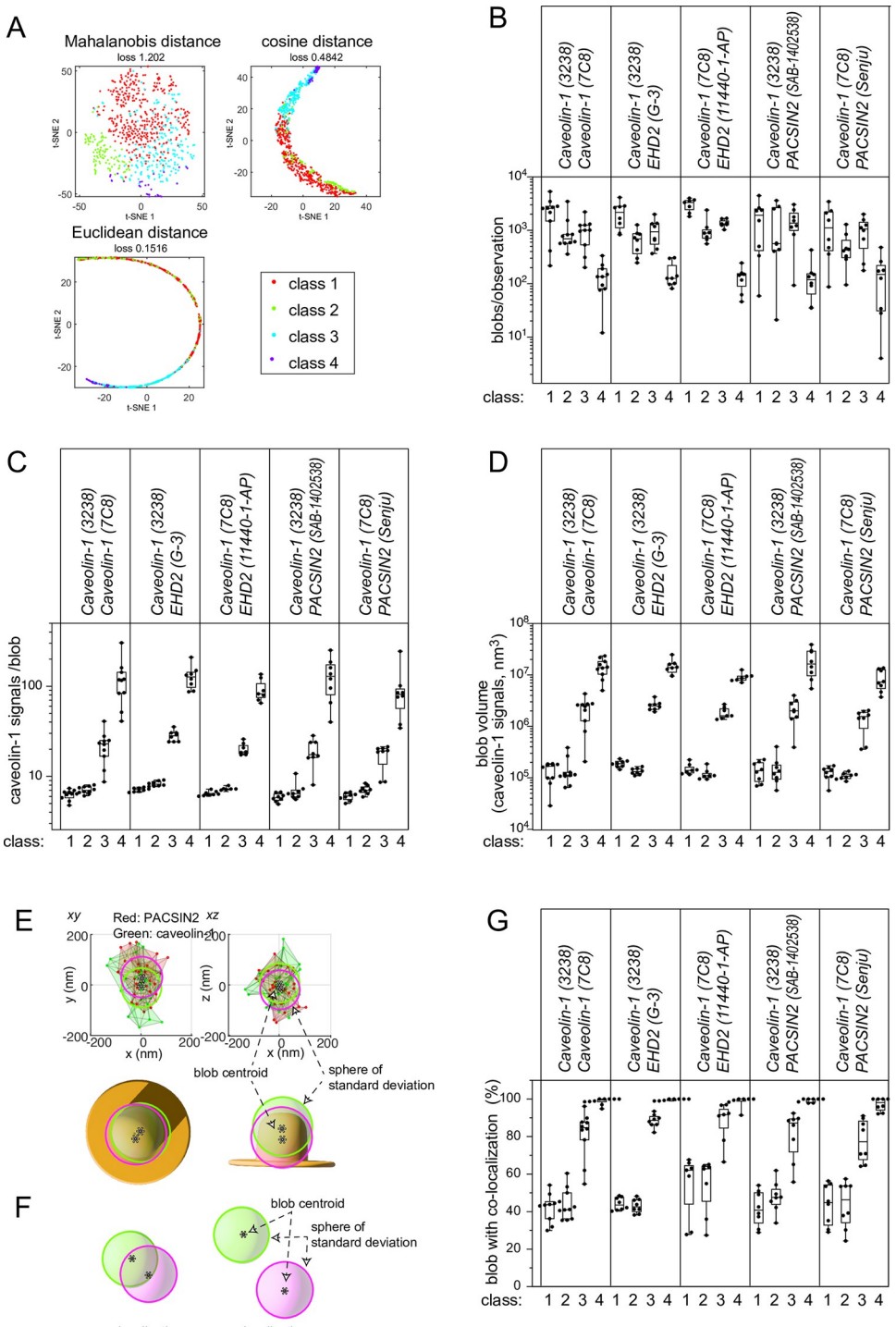

**Fig 2. Clustering of caveolin-1 into blobs and co-localization with PACSIN2 or EHD2.** (A) Typical clustering results of caveolin-1 signals of blobs from a cell into groups according to proximity by SuperResNet and their projections by t-SNE with Mahalanobis distance, cosine distance, and Euclidean distance. (B) The number of each class of caveolin-1 blobs per observation. (C) The average number of caveolin-1 signals from a blob of each class per observation. (D) The average volume of each class of caveolin-1 blobs per observation. In (B-D), the dot represents an average from an observation, which typically contained a cell. N = 6–10 observations for each combination of antibodies. (E) A caveolin-1 blob containing PACSIN2. Each signal of PACSIN2 and caveolin-1 is shown as a dot, and the signals within 80 nm of each other are connected by lines. The centroids of the caveolin-1 and PACSIN2 signals are illustrated by *, with spheres based on the radii of the standard deviations of caveolin-1 and PACSIN2. (F) The

definition of colocalization by the overlap of the spheres of standard deviations. (G) Percentages of the colocalizations of blobs. The percentages of caveolin-1 blobs colocalized with PACSIN2 and EHD2 are shown for each antibody combination. The co-staining of two caveolin-1 antibodies indicated the blobs with two co-localized caveolin-1 antibodies.

did not appear to be largely different from those of caveolin-1. Therefore, PACSIN2, EHD2, and caveolin-1 had largely overlapped spatial localizations in caveolin-1 blobs.

Next, we assessed the relative differences in the depth distributions by the distance of the centroid of caveolin-1 signals of the class 3 blobs to that of the associated PACSIN2 and EHD2 blobs. The negative distance in the z direction indicated a localization closer to the plasma membrane, i.e., the glass on which the cells were attached. The distances of the EHD2 centroids by the two kinds of antibodies to that of caveolin-1 were similar to each other and also to the centroid determined with another caveolin-1 antibody, indicating that the EHD2 localization was similar to that of caveolin-1 (Fig 3C). The monoclonal PACSIN2 antibody (SAB-1402538) recognizes the region before the SH3 domain of PACSIN2 that interacts with EHD2 (Fig 3D) [39]. The PACSIN2 centroid determined with the monoclonal antibody did not have a significant difference in depth from the EHD2 and caveolin-1 centroids (Fig 3C), which is consistent with the interaction of the region before the SH3 domain of PACSIN2 with EHD2. The polyclonal PACSIN2 antibody (Senju) recognizes the F-BAR domain, which binds to membrane [34]. Interestingly, the PACSIN2 centroid identified by this polyclonal antibody to the F-BAR domain was 5–10 nm closer to the plasma membrane than the EHD2 centroid (Fig 3C). The F-BAR domain of PACSIN2 is an arc-like rod of ~20 nm in length with a width of 5 nm [52]. The SH3 domains are globular domains with a diameter of ~5 nm [53], and EHD2 is also a globular protein of ~10 nm diameter [54]. Therefore, the PACSIN2 F-BAR domain was localized closer to the plasma membrane at a one protein distance, suggesting the role of PACSIN2 in connecting caveolae to the plasma membrane. Combined with the overall overlaps of PACSIN2, EHD2, and caveolin-1 localizations, these proteins would form a unit of PACSIN2, EHD2, and caveolin-1, in which the F-BAR domain of PACSIN2 faces the plasma membrane (Fig 3E).

This SMLM analysis suggested that PACSIN2 and EHD2 are localized in typical caveolae. We previously reported that PACSIN2 is localized to caveolae throughout the caveolar life cycle [32–34]. However, PACSIN2 appeared to tubulate membrane upon endocytosis and cholesterol depletion. The TIRF analysis indicated that PACSIN2 and caveolin-1 were co-localized, from the appearance of caveolin-1 at the plasma membrane, and the colocalization with dynamin occurred only at the last moment of endocytosis [40]. Upon cholesterol removal, PACSIN2 binding to the membrane was strengthened, supporting the production of membrane tubules, presumably for endocytosis [36]. Accordingly, caveolae with abundant cholesterol would exhibit a weaker affinity to PACSIN2, to prevent the elongation of caveolae for endocytosis. Therefore, the localizations of PACSIN2 and EHD2 at the entire caveola, with the exposure of PACSIN2 at the putative neck region, a boundary between the plasma membrane and the caveolar main body, would provide the potential for the extension of the caveola to such tubular structures upon endocytosis and cholesterol depletion.

## Materials and methods

### HeLa cell culture

HeLa cells were cultured as described previously [34] in Dulbecco's modified Eagle's medium (Nacalai), supplemented with 10% fetal calf serum, 63 μg/ml benzylpenicillin potassium, and 100 μg/ml streptomycin.

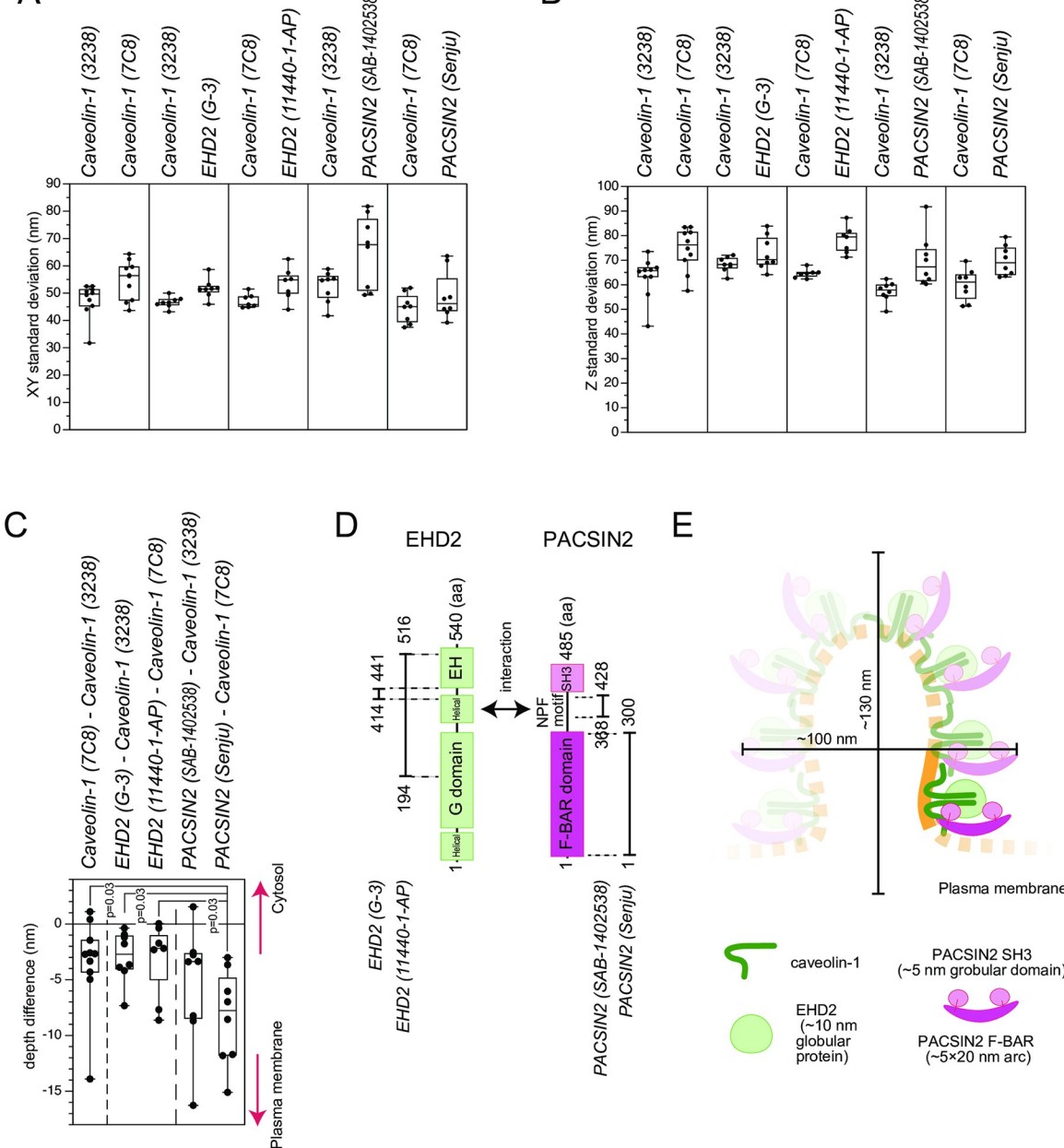

**Fig 3. The geometry of colocalization in the blobs.** (A) Standard deviations of the blobs of each antibody staining in the XY plane, the plane parallel to the lens surface or coverslip of the cell attachment; i.e., the plasma membrane. (B) Standard deviations of the blobs of each antibody staining in the Z or depth direction. In (A, B), the pairs of antibodies used for co-staining are shown side-by-side. The dot represents an average from an observation, which typically contained a cell. N = 6–10 observations for each combination of antibodies. (C) The difference in the blob centroid depths of the indicated antibodies used for co-staining. A negative value indicates closer localization to the plasma membrane. The dot represents an average from an observation, which typically contained one cell. N = 6–10 observations for each combination of antibodies. The statistical difference in the depth of the PACSIN2 antibodies was evaluated by one-way ANOVA with post-hoc Holm-Bonferroni analysis and the p-values were indicated. (D) Domain diagrams of PACSIN2 and EHD2. The antigens for the antibodies, as well as the interaction sites between PACSIN2 and EHD2, are also illustrated. (E) A model of PACSIN2 F-BAR domain localization in caveolae. The PACSIN2 F-BAR domain, SH3 domain, EHD2, and caveolin-1 are illustrated in an approximate scale. The size of the caveola is obtained from ~2 times of the standard deviations. The PACSIN2 F-BAR domain is located close to the plasma membrane, but PACSIN2, EHD2, and caveolin-1 mostly overlap each other, resulting in the hypothetical unit of PACSIN2, EHD2, and presumably caveolin-1 in caveolae. The shaded part is hypothetical.

## Antibodies

The anti-PACSIN2 rabbit polyclonal antibody (Senju) was affinity-purified from the serum of rabbits immunized with the F-BAR domain of PACSIN2 [34]. The anti-EHD2 rabbit poly-clonal antibody (11440-1-AP, Proteintech), the mouse monoclonal anti-caveolin-1 (7C8, Santa Cruz Biotechnology, sc-53564, 1:100), the rabbit polyclonal anti-caveolin-1 (Cell Signaling, #3238, 1:200), the mouse monoclonal anti-PACSIN2 (Sigma, SAB-1402538, 1:100), and the mouse monoclonal anti-EHD2 (G-3, Santa Cruz, sc-515458, 1:100) antibodies were purchased.

## STORM observation and analysis

The three-dimensional STORM setup with a cylindrical lens for depth measurement was pur-chased from Nikon and modified based on previous reports [42,44,45]. Dye preparation, sec-ondary antibody labeling, and cell staining for STORM imaging (Nikon) were performed according to the manufacturer's protocols, using combinations of Alexa Fluor 405 + Alexa Fluor 647 or Cy3 + Alexa Fluor 647 [44,45]. Alexa Fluor 405 and Cy3 are the activator dyes, and Alexa Fluor 647 is the reporter dye. The ratio of the activator dye: the reporter dye: anti-body is 2–3: 0.6–1: 1. HeLa cells were cultured on Lab-Tek II chambered cover glasses (Nunc) that were pre-cleaned with 1M KOH for 1 hr. They were fixed in 3% paraformaldehyde (WAKO) + 0.1% glutaraldehyde (TAAB, electron microscopy grade) in HEPES-buffered saline (30 mM Hepes, pH 7.4, 100 mM NaCl, 2 mM $CaCl_2$) for 10 min at room temperature, reduced with 0.1% $NaBH_4$ in PBS for 7 min, and blocked in blocking buffer (3% BSA + 0.2% Triton X-100 in PBS) for 1 hr at room temperature. The cells were then stained with a 1:100 dilution of the primary antibodies in the blocking buffer for at least 1 hr at room temperature. After wash-ing with wash buffer (0.05% Triton X-100, 0.2% BSA in PBS), the cells were incubated with secondary antibodies for 1 hr at room temperature and then washed. Finally, the cells were post-fixed with 3% paraformaldehyde + 1% glutaraldehyde in HEPES-buffered saline for 10 min at room temperature, and then stored in PBS at 4˚C.

For image acquisition, the cells were soaked in 50 mM Tris-HCl (pH 8.0), 10 mM NaCl, and 10% glucose supplemented with cystamine, glucose oxidase, and catalase, according to the manufacturer's instructions. An N-STORM (Nikon) super-resolution microscope equipped with a 100×/1.49 objective lens (Apo TIRF 100× Oil DIC N2, Nikon) and an EMCCD camera (iXon Du-897, ANDOR) was used for imaging. One image for the activation laser (405 or 561 nm) and three sequential images for the reporter laser (647 nm) were obtained for 10,000 cycles (total 40,000 images) and analyzed with the NIS-Elements AR 4.60.00 software provided by Nikon.

The signals upon the reporter laser irradiation following the observation by the activation laser irradiation were considered to be the specific signals and analyzed further. The coordi-nates that were at almost identical positions (<20 nm) in the continuous observations were eliminated, because the Alexa dye emits signals multiple times [55–57]. The coordinates were converted into the VISP format by the ChriSTORM ImageJ plugin [58–60], as described in the Supplementary Data 1 and 2. In all figures, each dot represents one signal. The clustering of the caveolin-1 signals was performed by SuperResNet [46,47], where >4 caveolin-1 signals within 80 nm, a value determined as a 20% smaller size of caveolin-1, were connected to a clus-ter. The clusters were divided into 4 classes according to the SuperResNet analysis of caveolin-1 [46,47], by the numbers of signals, shape parameters, and so on. Typically, thousands of blobs were identified per cell. The clusters of caveolin-1 signals exhibited blob structures. The volume of the class 3 blob corresponded to the volume of a sphere with a 60–100 nm diameter, a typical size of a caveola, which ranges from $0.9–4 \times 10^6$ $nm^3$. The PACSIN2 and EHD2

signals were also grouped into clusters by SuperResNet. The overlaps of the signal distribution of the signals were considered for co-localization; i.e., the centroid distance below the sum of each standard deviation (Fig 2F). The center of mass of the clusters was considered for the depth difference. These calculations were performed with MATLAB. For each blob, the standard deviations in the XY and Z axes and the depth differences between the centroids of clusters of each antibody labeling were examined and then averaged per each observation, which typically contains a cell. The averages of the observations were then plotted in each Figure. In Fig 3C, the statistical evaluation was examined by One-way ANOVA with post-hoc Holm-Bonferroni analysis.

## Supporting information

**S1 Fig. The statistics of blobs.** The average number of caveolin-1 blobs per observation, the XY standard deviation, and the Z standard deviation for each class and each combination of antibodies are shown. The dot represents an average from an observation, which typically contained one cell. N = 6–10 observations for each combination of antibodies. The combinations of antibodies are as follows: a: Caveolin-1 (7C8) + Caveolin-1 (3238), b: Caveolin-1 (3238) + EHD2 (G-3), c: Caveolin-1 (7C8) + EHD2 (11440-1-AP), d: Caveolin-1 (3238) + PACSIN2 (SAB-1402538), and e: Caveolin-1 (7C8) + PACSIN2 (Senju).
(PDF)

**S2 Fig. The statistics of blobs with or without colocalization.** The volume of a caveolin-1 blob and the average number of caveolin-1 signals per blob per observation, obtained for each class for each combination of antibodies, shown with or without the colocalization of the two stains. The dot represents an average from an observation, which typically contained one cell. N = 6–10 observations for each combination of antibodies described in S1 Fig. w: Caveolin-1 blobs with the colocalization; wo: Caveolin-1 blobs without the colocalization.
(PDF)

**S1 Data. The compressed files of the original coordinates in the VISP files.**
(ZIP)

**S2 Data. The compressed files of the original coordinates in the VISP files and the inventory of the files.**
(ZIP)

## Acknowledgments

We thank the laboratory members for fruitful discussions.

## Author Contributions

**Conceptualization:** Shiro Suetsugu.

**Data curation:** Tamako Nishimura, Shiro Suetsugu.

**Formal analysis:** Tamako Nishimura, Shiro Suetsugu.

**Funding acquisition:** Shiro Suetsugu.

**Investigation:** Tamako Nishimura, Shiro Suetsugu.

**Methodology:** Tamako Nishimura, Shiro Suetsugu.

**Validation:** Tamako Nishimura, Shiro Suetsugu.

**Visualization:** Tamako Nishimura, Shiro Suetsugu.

**Writing – original draft:** Shiro Suetsugu.

**Writing – review & editing:** Tamako Nishimura, Shiro Suetsugu.

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
