## [Decision Letter · Decision Letter 0]

24 Jan 2022

PONE-D-21-39140Superresolution analysis of PACSIN2 and EHD2 at caveolaePLOS ONE

Dear Shiro,

Thank you for submitting your manuscript to PLOS ONE. After careful consideration, we feel that it has merit but does not fully meet PLOS ONE’s publication criteria as it currently stands. Therefore, we invite you to submit a revised version of the manuscript that addresses the points raised during the review process.

You will see from the verbatim comments of the reviewers below that while they found your work of potential interest they both raised significant criticims that need to be addressed before publication. The reviewers are particularly concerned by the quality of the manuscript and English literacy. Several experimental controls and explanations in Material and Methods are missing together with statistics validation, cryptic figure legends and so on. 

I would therefore recommend to fully address the criticism of both reviewers who are each recognized experts in caveolae. I hope you will find these commenst useful and lokk forward to reading an improved manuscript in due time.

Best regards

Christophe Lamaze

We look forward to receiving your revised manuscript.

Kind regards,

Christophe Lamaze

Academic Editor

PLOS ONE

Journal Requirements:

Reviewers' comments:

Reviewer's Responses to Questions

**Comments to the Author**

1. Is the manuscript technically sound, and do the data support the conclusions?

Reviewer #1: Partly

Reviewer #2: Partly

2. Has the statistical analysis been performed appropriately and rigorously? 

Reviewer #1: Yes

Reviewer #2: I Don't Know

3. Have the authors made all data underlying the findings in their manuscript fully available?

Reviewer #1: Yes

Reviewer #2: Yes

4. Is the manuscript presented in an intelligible fashion and written in standard English?

Reviewer #1: Yes

Reviewer #2: Yes

5. Review Comments to the Author

Reviewer #1: The manuscript by Shiro Suetsugu entitled "SUperresolution analysis of PACSIN2 and EHD2 aat caveolae" concerns the super-resolution analysis of PACSIN2 and EHD2 localization in Hela cells. The author describes using STORM for single molecule localization of these two proteins relative to caveolin-1 used here as a marker for caveolae. The author compares the three-dimensional distribution of PACSIN2 and EHD2 in regard to the localization of caveolin-1. He shows that when classified into mature or immature blobs, half of the immature blobs contained EHD2 and PACSIN2 while most of the mature blobs had EHD2 and PACSIN2 suggesting a progressive association of these two proteins with maturation of caveolae. The author also shows that PACSIN2 then EHD2 were closer to the membrane compared to caveolin1 suggesting that they are localized at the neck of caveole.

Overall the idea to analyze PACSIN2 and EHD2 by STORM is interesting and the work is of good quality. My enthusiasm is slightly dimished because there are hardly any pictures to validate the findings and the paper is very minimalist. In fig.1 the individual channels could at least be shown separately. What about using Cavin proteins or other markers as controls to validate the findings? or using expression of the same tagged proteins to validate the results with the antibodies?

The parameters used to separate the different objects in SuperResNet are not described in detail in the methods section and it is difficult to understand how the clustering was performed. The author separates the objects into different classes and make assumptions on the degree of maturation but this could not be the case. One could imagine easily that caveolae could be more heterogeneous and some more or less flattened caveolae could still be mature. The term "maturity" is not appropriate and it would better to use comparison in terms of curvature as it is directly connected to the height of the objects which is what is being best measured by STORM.

The author chose to group PACSIN2 and EHD2 into clusters that were +/- 80nm to be associated with the caveolin1 blob. How can the authors then claim that these objects could be localized at the neck if they are so far from the caveolin signal? DO any of the images suggest accumulation of EHD2 or PACSIN2 at the neck? maybe the author could show an example of STORM image to substantiate this finding.

The authors mention on p5 that caveolae typical diameter is 100 nm while in reality it is 60-70nm at the ultrastructural level

There are several typos throughout the MS (for example title on p5 "STROM observation", etc...) and the english is not always correct. Could be useful to have a native english person proof-read the manuscript.

Reviewer #2: In this short article, Dr Suetsugu sets out to establish the superresolution analysis of PACSIN2 and EHD2 at caveolae. EHD2 and PACSIN2 are in addition to the caveolins and cavins, the most recognised and studied caveolar proteins. The research topic is interesting and topical, but the manuscript appears at places hastily put together and lacks some necessary controls.

Major comments

1) Materials and methods section is not complete, and need to be substantially expanded in order to provide context to the study. As it is, the data presented are hard to interpret.

Some additional examples where the methods section need updating

“Under subtitle: “HeLa cell culture, transfection, and live imaging”

The section reads

141 HeLa cells were cultured as described previously [28] in Dulbecco’s modified Eagle’s 5 142 medium (DMEM), supplemented with 10% fetal calf serum (FCS).

There is no mention on transfection or live imaging. The manuscript does not use those techniques.

2) There are no controls for that the signal observed by the antibody labelling is specific. Super resolution is a very sensitive technique, and at such antibody validation under specific experimental conditions, including fixation etc is critical. Discussion, and analysis of the fixative used is necessary, as fixation is well known to change the morphology and preservation of caveolae. The fixative will likely also impact on epitope availability, which confounds the conclusions, including of the monoclonal CAV1 used. Is the CAV1 epitope available to the antibody throughout the caveolae bulb? A comparison with widely used and characterised poly clonal CAV1 might yield this necessary insight.

3) No details on the statistical tests carried out are available and the comparisons in the figure legends. Likewise N numbers should also be included, biological repeats etc.

4) The figure legends need more information, forinstance no information in Figure 1, on what the cell this is, it would also be helpful to have the images displayed in split and merged channels. In addition, to me the colour is magenta, and not red as stated above the figures.

5) Justification for the reason behind “where >4 caveolin-1 signals within 80 nm chosen”. And justification of why “clusters within 80 nm were considered to be the PACSIN2 or EHD2 clusters close to the Caveolin-1”. Results/conclusions might have been very? different if these distances had changed.

Minor comments

Figure 2, bulb, blob or blub are listed in the figure axis, are they meant to be the same?

Sub title, “STROM observation and analysis”, should be “STORM observation and analysis” In general there are rather widespread typo’s

Reference for caveosome is needed.

Reference for caveolae size is needed.

Line 163 The “with the provided NIKON Software” software needs to be named.

It would be helpful, if the IF images were shown in separate channels, as well as the merged. This is the commonly accepted way of representing dual colour images.

Line 66 correct “with the role of caveolae in caveolae formation [27, 28, 31].”

Therefore, PACSIN1 and EHD2 were suggested to be localized at the mature caveolae.

Is this Pacsin 2 instead of Pacsin1? There is as far I can see no data presented on PACSIN1

6. PLOS authors have the option to publish the peer review history of their article (what does this mean?). If published, this will include your full peer review and any attached files.

Reviewer #1: No

Reviewer #2: No

---

## [Author Response · Author response to Decision Letter 0]

2 May 2022

Q) –Reviewer #1: The manuscript by Shiro Suetsugu entitled "SUperresolution analysis of PACSIN2 and EHD2 at caveolae" concerns the superresolution analysis of PACSIN2 and EHD2 localization in Hela cells. The author describes using STORM for single-molecule localization of these two proteins relative to caveolin-1 used here as a marker for caveolae. The author compares the three-dimensional distribution of PACSIN2 and EHD2 in regard to the localization of caveolin-1. He shows that when classified into mature or immature blobs, half of the immature blobs contained EHD2 and PACSIN2 while most of the mature blobs had EHD2 and PACSIN2 suggesting a progressive association of these two proteins with maturation of caveolae. The author also shows that PACSIN2 then EHD2 were closer to the membrane compared to caveolin1 suggesting that they are localized at the neck of caveolae.

A) We appreciate your kind offer to revise our manuscript and extend the deadline. In this revised manuscript, we think we have experimentally addressed most of the concerns raised by the reviewers.

Q) Overall the idea to analyze PACSIN2 and EHD2 by STORM is interesting and the work is of good quality. My enthusiasm is slightly diminished because there are hardly any pictures to validate the findings and the paper is very minimalist. In fig.1 the individual channels could at least be shown separately. What about using Cavin proteins or other markers as controls to validate the findings? or using expression of the same tagged proteins to validate the results with the antibodies?

A) Thank you for your valuable comments. According to your advice, we have separated the merged picture into the individual channels, as shown in the new Fig. 1. Also, we replaced Figure 1 with those representing the analysis of the STORM signal coordinates of the class 3 blobs.

To validate the caveolin-1 localization, we tested the two anti-cavin-1 antibodies (CST #69036 and BD #611258). However, they did not work under the conditions that were used to stain the other proteins, and therefore, the cavin-1 antibody was not used to validate the caveolin-1 staining. However, we tested the co-immunostaining by using two caveolin-1 antibodies, the mouse monoclonal anti-caveolin-1 (7C8) antibody (used in the previous version of the manuscript) and the rabbit polyclonal anti-caveolin-1 (3238) antibody. These antibodies both detect the N-terminal cytoplasmic portion of caveolin-1. The co-localization criteria were set to the overlaps of the spatial distributions of the signals, as shown in the revised Figure 2F. The class 3 blobs, which had ~20 caveolin-1 signals and typical caveolar volume, exhibited good co-localization (Figure 2G). However, the overall co-localization of these caveolin-1 antibodies was only ~50% in the class 1 and class 2 blobs, which have ~8 caveolin-1 signals. When we compared the numbers of signals of caveolin-1 and the volumes of the blobs, the number of signals and the volumes of the co-localized blobs were larger than those without colocalization (Figure S2), suggesting that class 1 and 2 blobs were too small to have co-staining or were not co-localized. For the antibody-dependent STORM signal, we confirmed the previous loss of caveolin-1 signals by the caveolin-1 siRNA, using the same caveolin-1 (7C8) antibody (Tachikawa et al., 2017). Therefore, we think the STORM observation of caveolin-1 was reconfirmed by the co-staining with two caveolin-1 antibodies for the blobs of the size of typical caveolae. 

We also tested the localizations of PACSIN2 and EHD2 by using mouse monoclonal antibodies in combination with polyclonal antibodies to caveolin-1, which is a different combination of antibodies from those in the previous manuscript; i.e., the epitopes are different. In addition, we observed PACSIN2 and EHD2 again with the antibodies used in the previous manuscript. On class 3 blobs, the polyclonal EHD2 antibody and monoclonal EHD2 antibody used in this study recognize overlapping regions of EHD2, and the localization of EHD2 relative to caveolin-1 was similar. In contrast, the monoclonal PACSIN2 antibody recognizes the linker region before the SH3 domain, which is close to the EHD2 binding site, whereas the polyclonal PACSIN2 antibody recognizes the F-BAR region. The position of the PACSIN2 antibody staining was ~5 nm different in depth relative to that of caveolin-1. Because two different antibodies to EHD2 gave similar localizations, the differential localizations of PACSIN2 antibodies were strongly suggested to result from the relative positions of PACSIN2.

Lastly, the antibody staining method in this paper is the combination of the activator and reporter dyes in the secondary antibody, for the detection of signals only when the activator signals were observed (Bates et al., 2007; Huang et al., 2008). The advantage of this method is the use of the same reporter dye to avoid chromatic aberration. The other methods using different wavelengths for the detection are not considered to be suitable for such analyses of co-localizations with nm accuracy. The use of fluorescent proteins requires two wavelengths for detection, and thus we did not use fluorescent proteins.

Q) The parameters used to separate the different objects in SuperResNet are not described in detail in the Methods section and it is difficult to understand how the clustering was performed. The author separates the objects into different classes and make assumptions on the degree of maturation but this could not be the case. One could imagine easily that caveolae could be more heterogeneous and some more or less flattened caveolae could still be mature. The term "maturity" is not appropriate and it would better to use comparison in terms of curvature as it is directly connected to the height of the objects which is what is being best measured by STORM.

A) In the revised manuscript, we described the details of the parameters. The signals were first filtered to cut the signals within 20 nm to avoid the possible blinking of the same antibody, which is also implemented in the NIKON software used for signal identification. The neighbor signals within 80 nm were labeled as clusters. The 80 nm value is the setting of the SuperResNet used in its publication (Khater et al., 2018), which also analyzed caveolae. We adapted these values because they are reasonable considering the sizes of caveolae, typically 60-70 nm in diameter. Furthermore, the number of observations of caveolin-1 was 4-15 with this antibody labeling according to our previous estimations (Tachikawa et al., 2017), and thus 80 nm for clustering appeared to be reasonable.

The four-class separation of clusters was also adopted from the paper (Khater et al., 2018) because it appears to extract the "mature" caveolae with ~20 caveolin-1 signals. However, according to the comments, we rephased the term maturation to the caveolin-1 signal abundance and the volumes of the blobs. 

We also measured the spatial distribution of caveolin-1 signals in the XY plane and Z (depth) direction by calculating the standard deviation of the coordinates of each blob. The standard deviation is considered to correlated with the radius of the blob. The class 1 and class 2 caveolin-1 clusters had 30-40 nm standard deviations in both XY and Z, and the class 3 clusters had 60-70 nm standard deviations in both XY and Z. Therefore, it was difficult to discuss the curvature of the blobs for the flatness of caveolae. This might indicate that there are few "flat" caveolae under our culture conditions of DMEM supplemented with serum.

Q) The author chose to group PACSIN2 and EHD2 into clusters that were +/- 80nm to be associated with the caveolin1 blob. How can the authors then claim that these objects could be localized at the neck if they are so far from the caveolin signal? DO any of the images suggest accumulation of EHD2 or PACSIN2 at the neck? maybe the author could show an example of STORM image to substantiate this finding.

A) The 80 nm value was adopted because the clustering threshold is also 80 nm for caveolin-1. However, according to your comments, it would not be appropriate to use it to connect the two labelings. Therefore, we recalculated the co-localization by the overlap of the spatial distribution of the signals, as examined by the standard deviation (Figure 2F).

Our previous Figure 3 showed typical images of the blobs shown by the lines connecting each signal, and were not easy to discern. We removed the Figure and replaced it with a cartoon for the measurement of the centroid and the standard deviation for the co-localization in the revised Figures 2E and F.

The word "neck" might be too exaggerated because PACSIN2 is just localized at ~5 nm, which is the size of one protein. Therefore, we modified the description to state that caveolin-1, EHD2, and PACSIN2 co-localize at class 2 caveolin-1 blobs, and that PACSIN2, especially its F-BAR domain, is thought to exist closer to the plasma membrane.

Q) The authors mention on p5 that caveolae typical diameter is 100 nm while in reality it is 60-70nm at the ultrastructural level

A) Thanks for your indication. Various papers have described the diameter as 70-100 nm, 50-100 nm, etc. We rephrased the diameter to 60-100 nm. 

Page 8： The volume of the class 3 blob corresponded to the volume of a sphere with a 60-100 nm diameter, a typical size of a caveola, which ranges from 0.9-4 × 106 nm3. 

Q) There are several typos throughout the MS (for example title on p5 "STROM observation", etc...) and the english is not always correct. Could be useful to have a native english person proof-read the manuscript.

A) We have corrected the typo as suggested and had an English proof-reading.

References

Fujimoto, T., Kogo, H., Nomura, R., and Une, T. (2000). Isoforms of caveolin-1 and caveolar structure. Journal of cell science 113, 3509-3517.

Huang, B., Wang, W., Bates, M., and Zhuang, X. (2008). Three-dimensional super-resolution imaging by stochastic optical reconstruction microscopy. Science 319, 810-813. 10.1126/science.1153529.

Khater, I.M., Meng, F., Wong, T.H., Nabi, I.R., and Hamarneh, G. (2018). Super Resolution Network Analysis Defines the Molecular Architecture of Caveolae and Caveolin-1 Scaffolds. Sci Rep 8, 9009. 10.1038/s41598-018-27216-4.

Sinha, B., Köster, D., Ruez, R., Gonnord, P., Bastiani, M., Abankwa, D., Stan, R.V., Butler-Browne, G., Vedie, B., Johannes, L., et al. (2011). Cells respond to mechanical stress by rapid disassembly of caveolae. Cell 144, 402-413. 10.1016/j.cell.2010.12.031.

Tachikawa, M., Morone, N., Senju, Y., Sugiura, T., Hanawa-Suetsugu, K., Mochizuki, A., and Suetsugu, S. (2017). Measurement of caveolin-1 densities in the cell membrane for quantification of caveolar deformation after exposure to hypotonic membrane tension. Sci Rep 7, 7794. 10.1038/s41598-017-08259-5.

Q) Reviewer #2: In this short article, Dr Suetsugu sets out to establish the superresolution analysis of PACSIN2 and EHD2 at caveolae. EHD2 and PACSIN2 are in addition to the caveolins and cavins, the most recognised and studied caveolar proteins. The research topic is interesting and topical, but the manuscript appears at places hastily put together and lacks some necessary controls.

A) We appreciate your kind offer to revise our manuscript and extend the deadline. In this revised manuscript, we think we have experimentally addressed most of the concerns.

Major comments

Q) 1) Materials and methods section is not complete, and need to be substantially expanded in order to provide context to the study. As it is, the data presented are hard to interpret.

Some additional examples where the methods section need updating

"Under subtitle: "HeLa cell culture, transfection, and live imaging"

The section reads

141 HeLa cells were cultured as described previously [28] in Dulbecco's modified Eagle's 5 142 medium (DMEM), supplemented with 10% fetal calf serum (FCS).

There is no mention on transfection or live imaging. The manuscript does not use those techniques.

We appreciate your helpful comments. We deleted these sentences because we did not transfect cells or perform live imaging. We also added more details in the methods.

Q) 2) There are no controls for that the signal observed by the antibody labeling is specific. Super resolution is a very sensitive technique, and at such antibody validation under specific experimental conditions, including fixation etc is critical. Discussion, and analysis of the fixative used is necessary, as fixation is well known to change the morphology and preservation of caveolae. The fixative will likely also impact on epitope availability, which confounds the conclusions, including of the monoclonal CAV1 used. Is the CAV1 epitope available to the antibody throughout the caveolae bulb? A comparison with widely used and characterised poly clonal CAV1 might yield this necessary insight.

A) Thank you for your kind advice. We think our fixation conditions (3% paraformaldehyde + 0.1 % glutaraldehyde) are similar to or the same as those in publications on caveolae, including ours (Tachikawa et al., 2017). Therefore, we tested other antibodies to confirm our results. We tested the co-immunostaining by using two caveolin-1 antibodies, the mouse monoclonal anti-caveolin-1 (7C8)antibody used in the previous version of the manuscript and the rabbit polyclonal anti-caveolin-1 (3238) antibody, which both detect the N-terminal cytoplasmic portion of caveolin-1. These antibodies co-localized with the class 3 blobs, with ~20 caveolin-1 signals, where the co-localization criteria were set to be the overlap of the spatial distribution of the signals, as shown in Figure 2F. The overall co-localization of these caveolin-1 antibodies was only ~50% with the class 1 and class 2 blobs, which have ~8 caveolin-1 signals on average. When we compared the numbers of signals of caveolin-1 and the volumes of the blobs, these values were larger than those without colocalization (Figure S2). Therefore, the co-staining appears to reflect the number of caveolin-1 molecules and the volume of the structure. The class 3 blobs, which had ~20 caveolin-1 signals, exhibited good co-localization (Figure 2G). For the antibody-dependent STORM signal, we have reconfirmed the loss of caveolin-1 signals by the caveolin-1 siRNA by using the same caveolin-1(7C8) antibody (Tachikawa et al., 2017). Therefore, we think the STORM observation of caveolin-1 was again confirmed by the co-staining of two caveolin-1 antibodies. 

We also tested the localizations of PACSIN2 and EHD2 by using mouse monoclonal antibodies in combination with the polyclonal antibodies to caveolin-1 on the class 3 blobs. This is a different combination of antibodies from those in the previous manuscript; i.e., the epitopes are different. We again observed PACSIN2 and EHD2, using the antibodies as in the previous manuscript. The polyclonal EHD2 antibody and monoclonal EHD2 antibody used in this study recognize the overlapping region of EHD2, and the relative localization of EHD2 to caveolin-1 was similar. In contrast, the monoclonal PACSIN2 antibody recognizes the linker region before the SH3 domain, which is close to the EHD2 binding site, whereas the polyclonal PACSIN2 antibody recognizes the F-BAR region. The relative position of PACSIN2 antibody staining was ~5 nm different in depth relative to caveolin-1. Because two different antibodies to EHD2 revealed similar localizations, the differential localization of PACSIN2 antibodies was strongly suggested to result from the relative positions of PACSIN2.

Q) 3) No details on the statistical tests carried out are available and the comparisons in the figure legends. Likewise N numbers should also be included, biological repeats etc.

A) We added more details in the Materials and Methods, as well as the Figure legends. For each blob, the standard deviations on the XY and Z axes and the depth difference between the centroid of the clusters of each labeled antibody labeling were examined and then averaged per observation, which typically contained one cell. The averages of the observations were plotted in each Figure, and were used for statistical evaluations by the Student's t-test.

Q) 4) The figure legends need more information, for instance no information in Figure 1, on what the cell this is, it would also be helpful to have the images displayed in split and merged channels. In addition, to me the colour is magenta, and not red as stated above the figures.

A) We improved the figure legends. The Hela cells are displayed with each channel image in the revised manuscript.

Q) 5) Justification for the reason behind "where >4 caveolin-1 signals within 80 nm chosen". And justification of why "clusters within 80 nm were considered to be the PACSIN2 or EHD2 clusters close to the Caveolin-1". Results/conclusions might have been very? different if these distances had changed.

A) The 80 nm value is the SuperResNet setting used in its publication (Khater et al., 2018), which also analyzed caveolae. We adapted these values because they are reasonable when considering the size of caveolae, typically 60-70 nm in diameter. Furthermore, the numbers of observations of caveolin-1 were 4-15 with this antibody labeling, according to our previous estimations (Tachikawa et al., 2017), and thus 80 nm for clustering appeared to be reasonable. This assumption is consistent with the labeling efficiency of the antibody, reported as 1-8 signals per caveola by electron microscopic analysis (Fujimoto et al., 2000).

The 80 nm value for the co-localizations was adopted because the clustering threshold is also 80 nm for caveolin-1. However, according to your comments, it would not be appropriate to use for connecting the two labels. Therefore, we recalculated the co-localization by using the criteria of the overlaps of the spatial distributions of the signals by standard deviations, as shown in the revised Figure 2F.

Minor comments

Q) Figure 2, bulb, blob or blub are listed in the figure axis, are they meant to be the same?

A) We have corrected these as "blob". We apologize for this mistake.

Q) Sub title, "STROM observation and analysis", should be "STORM observation and analysis" In general there are rather widespread typo's

A) We have corrected the word to "STORM".

Q) Reference for caveosome is needed.

A) We noticed that "caveolae rosette" is a more suitable word for describing the caveolae clusters on the plasma membrane, and adopted this phrase and added the references (Del Pozo et al., 2021; Sinha et al., 2011).

Q) Reference for caveolae size is needed.

A) We have added the references for the size, that are (Parton and Simons, 2007; Rothberg et al., 1992),

Q) Line 163 The "with the provided NIKON Software" software needs to be named.

A) We have added the name of the software.

Q) It would be helpful, if the IF images were shown in separate channels, as well as the merged. This is the commonly accepted way of representing dual colour images.

A) We have separated the merged picture into the individual channels, as shown in the new Fig. 1.

Q) Line 66 correct "with the role of caveolae in caveolae formation [27, 28, 31]."

A) We have replaced the first "caveolae" with "PACSIN2".

Q) Therefore, PACSIN1 and EHD2 were suggested to be localized at the mature caveolae.

Is this Pacsin 2 instead of Pacsin1? There is as far I can see no data presented on PACSIN1

A) As suggested, we have corrected the "PACSIN1" to "PACSIN2."

References

Bates, M., Huang, B., Dempsey, G.T., and Zhuang, X. (2007). Multicolor super-resolution imaging with photo-switchable fluorescent probes. Science 317, 1749-1753. 10.1126/science.1146598.

Del Pozo, M.A., Lolo, F.N., and Echarri, A. (2021). Caveolae: Mechanosensing and mechanotransduction devices linking membrane trafficking to mechanoadaptation. Curr Opin Cell Biol 68, 113-123. 10.1016/j.ceb.2020.10.008.

Fujimoto, T., Kogo, H., Nomura, R., and Une, T. (2000). Isoforms of caveolin-1 and caveolar structure. Journal of cell science 113, 3509-3517.

Huang, B., Wang, W., Bates, M., and Zhuang, X. (2008). Three-dimensional super-resolution imaging by stochastic optical reconstruction microscopy. Science 319, 810-813. 10.1126/science.1153529.

Khater, I.M., Meng, F., Wong, T.H., Nabi, I.R., and Hamarneh, G. (2018). Super Resolution Network Analysis Defines the Molecular Architecture of Caveolae and Caveolin-1 Scaffolds. Sci Rep 8, 9009. 10.1038/s41598-018-27216-4.

Parton, R.G., and Simons, K. (2007). The multiple faces of caveolae. Nature reviews. Molecular cell biology 8, 185-194. 10.1038/nrm2122.

Rothberg, K.G., Heuser, J.E., Donzell, W.C., Ying, Y.-S., Glenney, J.R., and Anderson, R.G.W. (1992). Caveolin, a protein component of caveolae membrane coats. Cell 68, 673-682. 10.1016/0092-8674(92)90143-Z.

Sinha, B., Köster, D., Ruez, R., Gonnord, P., Bastiani, M., Abankwa, D., Stan, R.V., Butler-Browne, G., Vedie, B., Johannes, L., et al. (2011). Cells respond to mechanical stress by rapid disassembly of caveolae. Cell 144, 402-413. 10.1016/j.cell.2010.12.031.

Tachikawa, M., Morone, N., Senju, Y., Sugiura, T., Hanawa-Suetsugu, K., Mochizuki, A., and Suetsugu, S. (2017). Measurement of caveolin-1 densities in the cell membrane for quantification of caveolar deformation after exposure to hypotonic membrane tension. Sci Rep 7, 7794. 10.1038/s41598-017-08259-5.

---

## [Decision Letter · Decision Letter 1]

1 Jun 2022

PONE-D-21-39140R1Super-resolution analysis of PACSIN2 and EHD2 at caveolaePLOS ONE

Dear Shiro,

Thank you for submitting your manuscript to PLOS ONE. After careful consideration, we feel that it should be published provided that you respond to the minor concerns raised by the two reviewers.  Therefore, we invite you to submit a revised version of the manuscript that addresses the minor points raised during the review process.

We look forward to receiving your revised manuscript.

Kind regards,

Christophe Lamaze

Academic Editor

PLOS ONE

Journal Requirements:

Reviewers' comments:

Reviewer's Responses to Questions

**Comments to the Author**

1. If the authors have adequately addressed your comments raised in a previous round of review and you feel that this manuscript is now acceptable for publication, you may indicate that here to bypass the “Comments to the Author” section, enter your conflict of interest statement in the “Confidential to Editor” section, and submit your "Accept" recommendation.

Reviewer #1: All comments have been addressed

Reviewer #2: (No Response)

2. Is the manuscript technically sound, and do the data support the conclusions?

Reviewer #1: Yes

Reviewer #2: Partly

3. Has the statistical analysis been performed appropriately and rigorously? 

Reviewer #1: Yes

Reviewer #2: I Don't Know

4. Have the authors made all data underlying the findings in their manuscript fully available?

Reviewer #1: Yes

Reviewer #2: Yes

5. Is the manuscript presented in an intelligible fashion and written in standard English?

Reviewer #1: Yes

Reviewer #2: Yes

6. Review Comments to the Author

Reviewer #1: The authors have now significantly improved their manuscript. They have improved both the figures and the text and proof-read their MS.

Minor points:

the authors must simplify the presentation of Figure 3D, this panel is too complicated for the reader and also modify Figure 3E schematic as caveolae are never forming tubule-like structures but they have an omega shape.

Reviewer #2: The manuscript has greatly improved.

However, there are some issues that need to be discussed and corrected before publication.

1) There are some incorrect uses of references in the introduction.

“there are four cavin isoforms, and they are essential for caveolae”. Ref 11, only covers CAVIN1, the three original “cavin family papers” should be referenced here to be precise doi: 10.1083/jcb.200903053,doi: 10.1038/ncb1887, doi: 10.1038/emboj.2009.46. In addition, Professor Pilch’s original papers on the discovery of CAVIN1 functional importance are not cited, the authors could also consider to include these for the statement on CAVIN1.

Ref 26 is the original of EHD2 localisation to caveolae, so the authors should also include this when being discussed.

2) Needs detailed discussion on the limitations of using primary/secondary Abs to study the fine scale localisation of components within structures as small as caveolae.

3) Are student T-tests (and what type was used?) appropriate for these type of analysis? The authors need to explain the justification for this. The use of correct statistical test is critical.

Minor comments

Line 42 “quite similar” is a subjective term, and the authors could consider changing this ot be more specific.

Line 81 Change “However, Pacsin2 was” to “However, Pacsin2 is”

Line 87, no mention of what type of cells, this is first introduced in line 89, introduction should be in line 87.

Line 115, ref style for Khater, 2018 is not correct.

7. PLOS authors have the option to publish the peer review history of their article (what does this mean?). If published, this will include your full peer review and any attached files.

Reviewer #1: No

Reviewer #2: No

---

## [Author Response · Author response to Decision Letter 1]

4 Jun 2022

Reviewer #1: The authors have now significantly improved their manuscript. They have improved both the figures and the text and proof-read their MS.

A) We would like to appreciate your careful comments on the improvement of our manuscript.

Minor points:

the authors must simplify the presentation of Figure 3D, this panel is too complicated for the reader and also modify Figure 3E schematic as caveolae are never forming tubule-like structures but they have an omega shape.

A) We improved the Figure 3D for simplicity and we modified Figure 3E into the omega shaped one.

Reviewer #2: The manuscript has greatly improved.

However, there are some issues that need to be discussed and corrected before publication.

A) We would like to appreciate your careful comments on the improvement of our manuscript. We modified it further according to your comments.

1) There are some incorrect uses of references in the introduction.

“there are four cavin isoforms, and they are essential for caveolae”. Ref 11, only covers CAVIN1, the three original “cavin family papers” should be referenced here to be precise doi: 10.1083/jcb.200903053,doi: 10.1038/ncb1887, doi: 10.1038/emboj.2009.46. In addition, Professor Pilch’s original papers on the discovery of CAVIN1 functional importance are not cited, the authors could also consider to include these for the statement on CAVIN1.

Ref 26 is the original of EHD2 localisation to caveolae, so the authors should also include this when being discussed.

A) We appreciate for pointing these important papers. We cited these papers (Bastiani et al., 2009; Hansen et al., 2009; Liu and Pilch, 2008; McMahon et al., 2009) for CAVINs. The ref26 (now ref 32) is also at line 63 for the discussion of EHD2 localization.

2) Needs detailed discussion on the limitations of using primary/secondary Abs to study the fine scale localisation of components within structures as small as caveolae.

A) According to this suggestion, we added several sentences.

Line 132-138: Therefore, the number of molecules to be detected is related to the volume of the blobs and the number of observed signals, suggesting that the class 1 and 2 blobs might have an insufficient number of molecules for colocalization detection or might not have colocalization. Such difficulty in analyzing small structures would probably result from the antibody labeling, where only a small fraction of the proteins can be visualized, as has been shown that 1-8 signals per caveola by electron microscopic analysis by using various caveolin-1 antibodies (Fujimoto et al., 2000).

3) Are student T-tests (and what type was used?) appropriate for these type of analysis? The authors need to explain the justification for this. The use of correct statistical test is critical.

A) The comparison can be justified only between the caveolin-1 (7C8) antibody-treated groups for the PACSIN2 and EHD2 localizations by the two different antibodies. Therefore, we believe a two-tailed t-test is suitable. However, we also performed one-way ANOVA with post-hoc Holm-Bonferroni analysis, and p values are replaced.

Minor comments

Line 42 “quite similar” is a subjective term, and the authors could consider changing this ot be more specific.

A) It was rephrased to “almost identical” with citation.

Line 42: Caveolin exists as three isoforms, and the caveolin-1 and caveolin-3 amino acid sequences are almost identical (Tang et al., 1996; Way and Parton, 1995).

Line 81 Change “However, Pacsin2 was” to “However, Pacsin2 is”

A) This part is the summary of the results of this paper, and we would like to keep “was” here.

Line 87, no mention of what type of cells, this is first introduced in line 89, introduction should be in line 87.

A) We corrected it accordingly.

Line 89: First, we observed the single-molecule localizations in antibody-stained HeLa cells.

Line 115, ref style for Khater, 2018 is not correct.

A) We formatted it.

Bastiani, M., Liu, L., Hill, M.M., Jedrychowski, M.P., Nixon, S.J., Lo, H.P., Abankwa, D., Luetterforst, R., Fernandez-Rojo, M., Breen, M.R., et al. (2009). MURC/Cavin-4 and cavin family members form tissue-specific caveolar complexes. J Cell Biol 185, 1259-1273. 10.1083/jcb.200903053.

Fujimoto, T., Kogo, H., Nomura, R., and Une, T. (2000). Isoforms of caveolin-1 and caveolar structure. Journal of cell science 113, 3509-3517.

Hansen, C.G., Bright, N.A., Howard, G., and Nichols, B.J. (2009). SDPR induces membrane curvature and functions in the formation of caveolae. Nat Cell Biol 11, 807-814. 10.1038/ncb1887.

Liu, L., and Pilch, P.F. (2008). A critical role of cavin (polymerase I and transcript release factor) in caveolae formation and organization. J Biol Chem 283, 4314-4322. 10.1074/jbc.M707890200.

McMahon, K.A., Zajicek, H., Li, W.P., Peyton, M.J., Minna, J.D., Hernandez, V.J., Luby-Phelps, K., and Anderson, R.G. (2009). SRBC/cavin-3 is a caveolin adapter protein that regulates caveolae function. Embo j 28, 1001-1015. 10.1038/emboj.2009.46.

Tang, Z., Scherer, P.E., Okamoto, T., Song, K., Chu, C., Kohtz, D.S., Nishimoto, I., Lodish, H.F., and Lisanti, M.P. (1996). Molecular cloning of caveolin-3, a novel member of the caveolin gene family expressed predominantly in muscle. J Biol Chem 271, 2255-2261. 10.1074/jbc.271.4.2255.

Way, M., and Parton, R.G. (1995). M-caveolin, a muscle-specific caveolin-related protein. FEBS Lett 376, 108-112. 10.1016/0014-5793(95)01256-7.

---

## [Decision Letter · Decision Letter 2]

22 Jun 2022

Super-resolution analysis of PACSIN2 and EHD2 at caveolae

PONE-D-21-39140R2

Dear Shiro,

We’re pleased to inform you that your manuscript has been judged scientifically suitable for publication and will be formally accepted for publication once it meets all outstanding technical requirements. I would like to personally congratulate you for this much improved revised version and for this great work.

Kind regards,

Christophe Lamaze

Academic Editor

PLOS ONE

Additional Editor Comments (optional):

Reviewers' comments:

Reviewer's Responses to Questions

**Comments to the Author**

1. If the authors have adequately addressed your comments raised in a previous round of review and you feel that this manuscript is now acceptable for publication, you may indicate that here to bypass the “Comments to the Author” section, enter your conflict of interest statement in the “Confidential to Editor” section, and submit your "Accept" recommendation.

Reviewer #1: All comments have been addressed

2. Is the manuscript technically sound, and do the data support the conclusions?

Reviewer #1: Yes

3. Has the statistical analysis been performed appropriately and rigorously? 

Reviewer #1: Yes

4. Have the authors made all data underlying the findings in their manuscript fully available?

Reviewer #1: Yes

5. Is the manuscript presented in an intelligible fashion and written in standard English?

Reviewer #1: Yes

6. Review Comments to the Author

Reviewer #1: All good and ready to go! Congratulations to the authors on this much improved version describing super-resolution analysis of caveolae components.

7. PLOS authors have the option to publish the peer review history of their article (what does this mean?). If published, this will include your full peer review and any attached files.

Reviewer #1: No

---

## [Editor Report · Acceptance letter]

6 Jul 2022

PONE-D-21-39140R2 

Super-resolution analysis of PACSIN2 and EHD2 at caveolae 

Dear Dr. Suetsugu:

I'm pleased to inform you that your manuscript has been deemed suitable for publication in PLOS ONE. Congratulations! Your manuscript is now with our production department. 

Kind regards, 

on behalf of

Dr. Christophe Lamaze 

Academic Editor

PLOS ONE